# Study of Postural Stability Features by Using Kinect Depth Sensors to Assess Body Joint Coordination Patterns

**DOI:** 10.3390/s20051291

**Published:** 2020-02-27

**Authors:** Chin-Hsuan Liu, Posen Lee, Yen-Lin Chen, Chen-Wen Yen, Chao-Wei Yu

**Affiliations:** 1Department of Computer Science and Information Engineering, National Taipei University of Technology, Taipei 10608, Taiwan; chinhsuanliu@gmail.com (C.-H.L.); david741002@gmail.com (C.-W.Y.); 2Department of Occupational Therapy, I-Shou University, Kaohsiung 82445, Taiwan; posenlee@isu.edu.tw; 3Department of Mechanical and Electro-Mechanical Engineering, National Sun Yat-Sen University, Kaohsiung 80424, Taiwan

**Keywords:** depth sensors, standing still, postural control, inter-joint coordination, principal component analysis

## Abstract

A stable posture requires the coordination of multiple joints of the body. This coordination of the multiple joints of the human body to maintain a stable posture is a subject of research. The number of degrees of freedom (DOFs) of the human motor system is considerably larger than the DOFs required for posture balance. The manner of managing this redundancy by the central nervous system remains unclear. To understand this phenomenon, in this study, three local inter-joint coordination pattern (IJCP) features were introduced to characterize the strength, changing velocity, and complexity of the inter-joint couplings by computing the correlation coefficients between joint velocity signal pairs. In addition, for quantifying the complexity of IJCPs from a global perspective, another set of IJCP features was introduced by performing principal component analysis on all joint velocity signals. A Microsoft Kinect depth sensor was used to acquire the motion of 15 joints of the body. The efficacy of the proposed features was tested using the captured motions of two age groups (18–24 and 65–73 years) when standing still. With regard to the redundant DOFs of the joints of the body, the experimental results suggested that an inter-joint coordination strategy intermediate to that of the two extreme coordination modes of total joint dependence and independence is used by the body. In addition, comparative statistical results of the proposed features proved that aging increases the coupling strength, decreases the changing velocity, and reduces the complexity of the IJCPs. These results also suggested that with aging, the balance strategy tends to be more joint dependent. Because of the simplicity of the proposed features and the affordability of the easy-to-use Kinect depth sensor, such an assembly can be used to collect large amounts of data to explore the potential of the proposed features in assessing the performance of the human balance control system.

## 1. Introduction

Human beings are bipedal; therefore, balance in humans is extremely complex [1]. Human bodies are complex assemblies, which require continuous active control even when standing still. Such active control is achieved through appropriate spatial and temporal body segment coordination. Fatigue, diseases, injuries, and aging can compromise this control of balance [2].

Postural control is the ability to maintain equilibrium by maintaining or returning the center of body mass over its base of support and can be defined as the act of maintaining, achieving, or restoring a state of balance [3]. Studies have shown that a decline in postural control ability leads to a deficit in balance and increases the risk of falling [4,5,6,7,8]. Therefore, developing a simple assessment method for postural stability can predict the risk of falling. This is crucial because falls are a major public health problem [9,10]. According to the data published by the World Health Organization (WHO) in January 2018, approximately 646,000 individuals died from falls each year, and falls are the second leading cause of worldwide accidental injury deaths. Globally, the number of people requiring medical attention because of falls was estimated to be approximately 37,300,000 per year [11]. The WHO data also showed that elderly people are the most affected in fatal falls (over 65 years old). Approximately 28–35% of people aged 65 and over fall each year [12]. This probability increases to 32–42% for those over 70 years old. Therefore, developing measures that can characterize the effects of aging on postures can be very useful for fall risk assessment.

In performing a real-life task, the inherent degrees of freedom (DOFs) of the motor system of our body are typically considerably larger than the minimum DOFs required for performing the task. This is identified as the motor redundancy problem (also known as Bernstein’s problem) [13]. To understand the response of the human brain to Bernstein’s problem, several studies have examined the coordination of the finger forces when gripping. It was found that in some specific tasks, finger forces often exhibit a tendency of synchronization [14,15]. The degree of synchronization typically changes with the nature of the task [16,17] and the age of the subjects [18,19,20].

A fundamental problem of postural control is the coordination of multiple joints of the body to maintain postural stability. Because the number of freely movable body joints is considerably larger than the DOFs required for the postural balancing, the human balance control system clearly must cope with Bernstein’s problem. Many studies have suggested that postural balance is controlled by only one or a few joints. Among these studies, some have suggested that the ankle strategy or hip strategy or a combination of these two strategies is used in the human postural control system to maintain a static standing balance [21,22,23,24,25,26]. However, the validity of these simplified strategies has been questioned in recent studies. For example, the analysis of the effect of joint variation on the stability of the center of mass (COM) determined that almost all major joints are highly active when standing still [27]. Principal component analysis (PCA) was used to quantify the angular variation of the upper leg, lower leg, head, and upper limb. The results showed that the angle of the upper leg and trunk considerably affects the motion of the COM [28]. After measuring the kinematics of the ankle, knee, and hip joints by using an imaging system, a study determined that all leg joints play an influential role in maintaining the static standing balance [29]. In summary, these results clearly demonstrated the inadequacy of earlier studies, which assumed that only a few joints were actively involved in maintaining postural balance. However, coordination of the joints of the body to achieve a stable posture is a topic that is underresearched.

To study how the postural control system of the human body resolves Bernstein’s problem, the coordination patterns of the joint velocities were investigated. An extreme strategy for managing the redundancy is to minimize the DOFs of the postural control system to one by completely coupling all joint motions. The other extreme postural control strategy is to maximize the DOFs by making all joint motions perfectly independent and thus uncorrelated. In this study, we hypothesized that the human postural control system uses an approach that is an intermediate of two extreme strategies. To test this hypothesis, a Microsoft Kinect sensor was used. Several feature sets were used to characterize the coordination patterns of the joints of the body by studying how joint velocities interact with one another.

In addition to its simplicity and affordability, the reliability and validity of the Kinect sensor for human joint center measurements have been extensively tested and verified in many experimental studies [30,31,32,33,34,35,36,37,38]. Its measurement errors have also been carefully investigated [39,40,41,42,43]. These results clearly support the use of the Kinect sensor for the assessment of gait and balance performance [44,45,46,47,48,49,50,51,52,53,54,55,56]. It should be noted that many similar RGB-depth (RGB-D) sensor devices are already available. Interested readers are referred to a recent review paper for these Kinect alternatives [57]. We have also surveyed recent studies about the applications of depth camera for human motion capture and analysis. Based on the survey results, Table 1 summarizes the specifications of the most popular RGB-D sensors. Among all these popular RGB-D sensing devices, the Microsoft Kinect V2 sensor provides high-resolution color images, large field of view (FOV) areas, and powerful software development kits for the skeleton and joint detection. In addition, as previously addressed in this paper, the validity and reliability of the Kinect sensor have also been extensively studied and verified. These are the reasons why we chose Kinect V2 as our measurement device for joint tracking.

## 2. Materials

### 2.1. Kinect Depth Sensor System

The measurement system consisted of a Microsoft Kinect sensor connected to a personal computer based signal processing system. The Microsoft Kinect V2 sensor, also known as the Xbox One Kinect, provides five video related data streams which include color (1920 × 1080 @ 30 Hz), infrared (512 × 424 @ 30 Hz), depth (512 × 424 @ 30 Hz) images as well as body index (512 × 424 @ 30 Hz), and the skeleton information for every tracked person (25 joint centers @ 30 Hz). Note that the joint positions are provided at a resolution of 4 bytes per coordinate and hence 12 bytes per joint. The tracking volume of Kinect V2 is defined by the field of view (FOV, 70˚ horizontally, 60˚ vertically) and the range of depth-sensing (0.5–4.5 m). In this work, the Microsoft Kinect Software Development Kit (SDK) 2.0 was used to obtain the location of 25 human joint centers. By assuming the relative motions between the wrist, hand, and thumb centers of the same arm to be negligible when standing sill, hand and thumb joints were excluded from this study. Since ankles and feet are relatively motionless when standing still, these joints were also not included in this study. As a result, as shown in Figure 1, the 15 joints included in this study are the (1) head, (2) neck, (3) shoulder center, (4) left shoulder, (5) right shoulder, (6) trunk center, (7) left elbow, (8) right elbow, (9) hip center, (10) left hip, (11) right hip, (12) left hand, (13) right hand, (14) left knee, and (15) right knee. Note that “joint 1” represents the center of the head, which cannot perform a rotational movement, therefore “joint 1” is not rigorously a kinematic joint. Considering the velocity of the head center is different from the velocities of other body joints and the potential role of head movement on postural stability, this work included the head center in this study. However, to simplify the corresponding statements and discussions, this manuscript still refers the head center as “joint 1”.

As shown in Figure 2, the Kinect camera was placed approximately 2 m away in front of the subjects and was approximately 72–76 cm above the floor. We set up a green curtain at the back of the tested subjects to prevent possible interferences from the background.

### 2.2. Participants

Data were collected from 45 (15 youths, 30 elders) healthy participants who did not have a neurological or musculoskeletal impairment. The participants had no lower-limb discomfort and could maintain a double-leg stance with both eyes open. As mentioned previously, 15 youths (age: 24.06 ± 2.02 years old; body height: 174.20 ± 6.80 cm; body weight: 73.26 ± 15.21 kg; body mass index: 23.96 ± 3.79, respectively) and 30 elders (age: 71.13 ± 4.56 years old; body height: 162.03 ± 9.04 cm; body weight: 63.61 ± 10.32 kg; body mass index: 24.24 ± 3.53) participated in this study. Aging has been known to be associated with many postural stability impairing factors, each of which may have different impacts on inter-joint coordination patterns (IJCPs). In view of such uncertainties, the sample size of the older age group of this study was chosen to be larger than that of the younger age group.

### 2.3. Standing Still Experiment

Three 40-s test sessions were performed on each subject. In each session, the participants were instructed to look straight at a visual reference and standstill (with arms at the side) in a comfortable stance for 40 s. The distance between the visual reference and the test subject was about 2 m. The data collected from 5 to 35 s of the trials were used for this study. Every session was separated by approximately 1 minute of rest.

### 2.4. Data Processing

A data point of a joint center signal includes three coordinates (*x, y, z*), with *x, y, z* representing the mediolateral (ML), vertical, and anteroposterior (AP) directions, respectively. Only the ML direction was considered in the study for the following reasons. First, vertical direction data were not included because when standing still, the vertical direction movement is considerably smaller than those in the ML and AP directions. Second, AP direction data were not included since our preliminary experimental results suggest that the proposed features are relatively ineffective in dealing with the AP direction velocity signals. Finally, the most crucial reason is that ML balance impairments have been proven to be closely associated with falls in older people [58,59,60].

The signals of the joint center were recorded at a sampling rate of 30 Hz and filtered using a sixth-order zero-phase Butterworth filter with a cutoff frequency of 5 Hz. The joint velocity signals were then obtained using a five-point central difference method (five-point stencil). Finally, before computing the proposed features, the magnitudes of these joint velocity signals were normalized such that the energies for all the normalized joint velocity signals were equally large in each of the experimental trials.

## 3. Methodology

### 3.1. Local Inter-Joint Coordination Pattern (IJCP) Features

As shown in Figure 3, as a pre-processing step for feature generation, each of the 30-s joint velocity signals was divided into 30 one-sec nonoverlapping subintervals. With a 30 Hz sampling rate, each of such subintervals shown in Figure 3 consists of 30 sampling points. The temporal correlation coefficient *c_ij_*[*k*] is then specified as the Pearson product-moment correlation coefficient between the *k*th subinterval of joint *i* and the *k*th subinterval of joint *j* for *k* = 1, 2, …, 30.

Based on these temporal correlation coefficient signals of *c_ij_*[*k*], the following local IJCP feature sets were proposed:The mean of the temporal correlation signal *c_ij_*[*k*] was used for *k* = 1, 2, …, 30 to quantify the coupling strength between joints *i* and *j*. In the paper, these features are referred to as the coupling strength (CS) features.Considering the possibility that the inter-joint coupling behaviors can be time-varying, the second proposed feature set was used to characterize the changing speed of IJCPs. Specifically, for a given joint pair, by defining the zero-crossing point (ZCP) as the time instant that its correlation coefficient changes sign, the proposed IJCP changing speed feature is defined as the number of ZCPs and is referred as the ZCP feature, hereafter.Because the velocities of a pair of joints can be either positively or negatively correlated, the third set of features is defined as the ratio of the number of the negative *c_ij_*[*k*] to 30. This is because 30-time subintervals were conducted in each of our experimental trials. The negative correlation (NC) features were introduced to quantify the complexity of the pairwise IJCP.

As an analogy, the sign of a correlation coefficient can be compared to the two sides of a coin. The outcome of coin tosses is the most unpredictable and thus most complex when the coin is perfectly unbiased toward either side of the coin. If the coin becomes increasingly biased to either side, the outcome of the coin tosses becomes more predictable. Similarly, the joint coordination pattern of a joint pair can be considered most complex when the value of the NC feature is 0.5. When the value of an NC feature changes from 0.5 toward 1 or 0, the IJCP becomes less complex.

Mathematically, for a pair of joint velocity signals, its CS feature is defined as the mean of the corresponding temporal correlation coefficient signal, its ZCP feature is defined as the number of zero-crossing points of this temporal correlation coefficient signal and its NC feature is defined as the ratio of time that this temporal correlation signal has a negative value. In summary, this study proposes three sets of features to quantify the coupling strength (CS features), changing speed (ZCP features), and complexity (NC features) of the pairwise IJCPs, respectively. For joints *i* and *j*, these three types of features are denoted as *CS_ij_, ZCP_ij_,* and *NC_ij_,* respectively.

The features *CS_ij_*, *ZCP_ij_*, and *NC_ij_* characterize the coupling properties for joint pairs. To study the relative role of each joint in inter-joint coordination, this work extends the utility of these features by introducing three sets of joint-specific features. In specific, to extend CS features to a joint-specific level for joint *i*, we calculate the mean of *CS_ij_* for *j* = 1, 2, …, 15 and *j* ≠ *i*. By denoting this feature as JCS*_i_*, this joint-specific CS feature is used to quantify the overall coupling strength of joint *i*. In an identical manner, this work also extends the utility of ZCP and NC features and denote their joint-specific features as JZCP*i* and JNC*i*, respectively.

### 3.2. Global Inter-Joint Coordination Features

Features proposed in the previous subsection characterize the IJCP on a pairwise and thus local level. By contrast, a set of features from a global point of view is proposed in this subsection. The joint velocity vector *x*[*k*] associated with the *k*th sampling instant is defined as a 15-dimensional vector whose *i*th (*i* = 1, 2, …, 15) element is the velocity of the *i*th joint at the *k*th sampling instant. With a 30-s signal length and a sampling frequency of 30 Hz, 900 *x*[*k*] vectors were collected for each experimental trial. PCA was performed on these joint velocity vectors for each experimental trial, and the resulting eigenvalues and eigenvectors are denoted as *λ_i_* and ***v****_i_* for *i* = 1, 2, …, 15, respectively.

For the problem under consideration, eigenvectors represent a complete set of orthogonal modes of joint velocity movement. Because eigenvalue *λ_i_* represents the proportion of variance (PoV) explained by eigenvector ***v**_i_*, eigenvalues characterize the relative contributions of these 15 modes of joint velocity movement represented by the eigenvectors. Therefore, by using eigenvalues to quantify the contributions of each mode of joint velocity movement, PoV features were defined as *PoV_i_* = *λi* / (*λ*_1_ + *λ*_2_ +…+ *λ_15_*). In addition to these PoV features, the following PoV entropy feature was proposed to characterize the complexity of IJCPs from a global perspective:(1)$$E_{PoV}=-\sum_{i=1}^{15}PoV_{i}\log_{2}PoV_{i}$$

When joints are completely independent and thus uncorrelated, *PoV*_1_ = *PoV*_2_ =…= *PoV*_15_ = 1/15, the *E_PoV_* entropy feature has a maximum value of 3.907. By contrast, when all the joint velocities are perfectly correlated, the value of *PoV*_1_ is 1, and the values of the remaining PoV features are all zero. In this extreme case of total dependence, *E_PoV_* has a minimum value of 0. Based on such results, the PoV entropy feature *E_PoV_* was used to quantify the degree of overall inter-joint coupling complexity.

## 4. Results

In the first part of this section, the results obtained using the local IJCP features introduced in Section 3.1 are presented. Independent two-sided *t*-tests were performed to compare the means of the proposed features of the younger and older age groups in which a difference was considered significant when the *p*-value was less than 0.05. The following statistical results were obtained using the average values of the proposed features over three experimental trials for each tested subject. As a result, the sample sizes for the younger and older age groups were 15 and 30, respectively.

Because 15 joints were considered, 15 × 14/2 = 105 joint pairs were studied. In each of these 105 joint pairs, the CS feature means of the older age group were larger than the corresponding CS feature means of the younger group, that is, the mean value of *CS_ij_* of the older age group was larger than that of the younger age group for *i* = 1, 2, …, 15, *j* > *i*. Similarly, the results of the conducted experiments showed that the older age group has smaller *ZCP_ij_* and *NC_ij_* than those of the younger group for *i* = 1, 2, …, 15, *j* > *i*.

Among 105 comparative results of the CS features, 102 were significant. Among the 105 differences of the ZCP features, 98 were significant. Finally, among 105 sets of NC feature comparative results, 97 were significant. The histogram of the NC features is presented in Figure 4 for both age groups.

Figure 5 depicts the values of these joint-specific JCS*_i_* features for both age groups. The values of the joint-specific JZCP*_i_* features are plotted in Figure 6. Similarly, Figure 7 depicts the values of JNC*_i_*. The corresponding *p*-values of these statistical tests are summarized in Table 2. As shown in Table 2, only JCS_12_ (*p*-value = 0.07) and JNC_12_ (*p*-value = 0.097) yield nonsignificant results.

The results obtained using the global IJCP features introduced in Section 3.2 are presented in the second part of this section. Figure 8 depicts the proposed PoV features of the younger and older age groups. The means of the PoV features of both age groups and the corresponding *p*-values are summarized in Table 3. Finally, the mean and standard deviation of the PoV entropy feature are 2.602 and 0.403, respectively, for the younger group and 2.245 and 0.483, respectively, for the older group. The corresponding *p*-value was 3.75 × 10^−5^.

## 5. Discussion

As presented in the first part of Section 4, the values of the CS features of the older age group were larger than those of the younger age group. CS features quantifying the coupling strength of the joint pairs suggested that the inter-joint couplings of the older age group were larger than those of the younger group.

Furthermore, the results in the previous section demonstrated that the older age group has smaller ZCP values. This indicated that linear correlation coefficients of the younger group changes sign more rapidly than those of the older age group. This probably implies that the speed-of-response of the postural control system of the younger group was faster than that of the older age group. Thus, the postural control systems of the younger group can adapt more readily than those of the older age group to uncertainties such as disturbances, which can induce balance instability.

The results reported in Section 4 showed that the values of the NC features of the older age groups are smaller than those of the younger age group. As shown in Figure 4, all NC feature values were smaller than 0.5, which indicated that joint velocities are positively correlated most of the time. However, as shown in Figure 4, 23 of the younger age group’s NC features were larger than 0.4. By contrast, the values of the older age group NC features were smaller than 0.4. These results suggested that the pairwise inter-joint coupling of the older age group was less complex than that of the younger age group.

Regarding the joint-specific results, Figure 5 shows that JCS*_i_* of the older age group is larger than the corresponding JCS*_i_* of the younger group for all *i* (i.e., *i* = 1, 2, …, 15). Similarly, the joint-specific results depicted in Figure 6 and Figure 7 show that JZCP*_i_* and JNC*_i_* features of the older group are smaller than JZCP*_i_* and JNC*_i_* features of the younger group for all *i*. In addition, Figure 5 illustrates that the JCS*_i_* values of the older and younger age groups have very similar joint variation patterns. Specifically, the correlation coefficient between the two JCS curves of Figure 5 is 0.991. Similarly, the correlation coefficients between the two JZCP curves of Figure 6 and the two JNC curves of Figure 7 are 0.986 and 0.981, respectively. These results demonstrated that, when characterized by the proposed local IJCP features, the relative roles of the joints are age-independent.

Based on Figure 5, Figure 6 and Figure 7 and the relative locations with respect to the central axis of the body, the studied joints were categorized into two groups. The axial group consisted of the head (joint 1), neck (joint 2), shoulder center (joint 3), trunk center (joint 6), and hip joints (joints 9–11). The limb group consists of left shoulder (joint 4), right shoulder (joint 5), elbows (joints 7, 8), hands (joints 12, 13), and knees (joints 14, 15). The reason why for such a group division study is to investigate the potential associations between the relative locations of the joints with respect to central axis and the values of the proposed features in order to gain more insights about IJCPs. Compared with the joints of the axial groups, limb group joints have smaller JCS and larger JZCP and JNC values. These results suggested that some basic differences were observed among the IJCPs of these two joint groups. In addition, among all the joints being studied, the two knee joints (joints 14 and 15) have the two lowest joint-specific JCS values and the two highest JZCP and JNC joint-specific values. These results clearly demonstrated the unique role of the knee joints when standing still. By comparison, by having the largest JCS value and the lowest JZCP and JNC values, joint 6 (trunk center joint) exhibits the exact opposite properties.

As shown in Figure 8, the first PoV feature (i.e., *PoV*_1_) was considerably larger than the remaining PoV features for both age groups. This dominant role of *PoV*_1_ was more pronounced for the older age group. With the exception of *PoV*_1_, the values of *PoV_i_* of the older age group are smaller than those of the younger age group for *i* = 2, 3, …, 15. With the eigenvectors representing a complete set of orthogonal modes of joint velocity movement, the results shown in Figure 8 demonstrate that the relative contributions of these 15 modes of joint velocity movement of the younger age group are less skewed toward the first principal component, and thus more evenly distributed than the older age group. In addition, the considerably larger mean of the PoV entropy feature of the younger age group (2.602) than that of the older age group (2.245) suggested that the younger age group has more complex IJCPs than the older age group. In summary, these results supported the hypothesis of an approach intermediate to two extreme coordination modes of total joint dependence and independence is used in the inter-joint coordination strategy.

The results presented in this study cannot directly associate the proposed features with postural instability problems. However, considering the fact that postural instabilities increase significantly with age and the proposed features can very effectively detect the aging effects on IJCP, we believe that the potential of the proposed features in detecting postural instabilities warrant further study. In fact, although not presented in this work, our preliminary experimental results have shown that the results obtained by the proposed approach are in agreement with the conventional postural steadiness results obtained by force platform measurements.

## 6. Conclusions

The IJCPs, when standing still, were studied using Kinect to measure the motion of 15 body joints. Based on the proposed features, our results showed that aging increases the coupling strength, decreases the changing speed, and reduces the complexity of ICJPs. The results also supported the hypothesis that an inter-joint coordination strategy intermediate to total joint independence and joint dependence was used. In addition, the older age group tended more toward the total joint-dependence strategy than the younger age group.

The limitations of this study are as follows: First, the validity of the results can be further affirmed by increasing the number of participants. Second, only the ML direction motion was considered in this study. However, the results of this study are still valuable because many studies have suggested that lateral instability is a major cause of falling in the older population. The third limitation is the inability of the proposed approach in studying the role of the ankle joints because ankle joint centers are relatively motionless when standing still.

Considering its simplicity and affordability, the proposed approach can be used to collect large amounts of data to promote the development of effective predictive measures for falls. To further test the efficacy of the proposed features for detecting postural instabilities, a possible future study is to use the proposed features to characterize the effects of balance impairing factors such as sensorimotor deficits.

## Figures and Tables

**Figure 1 sensors-20-01291-f001:**
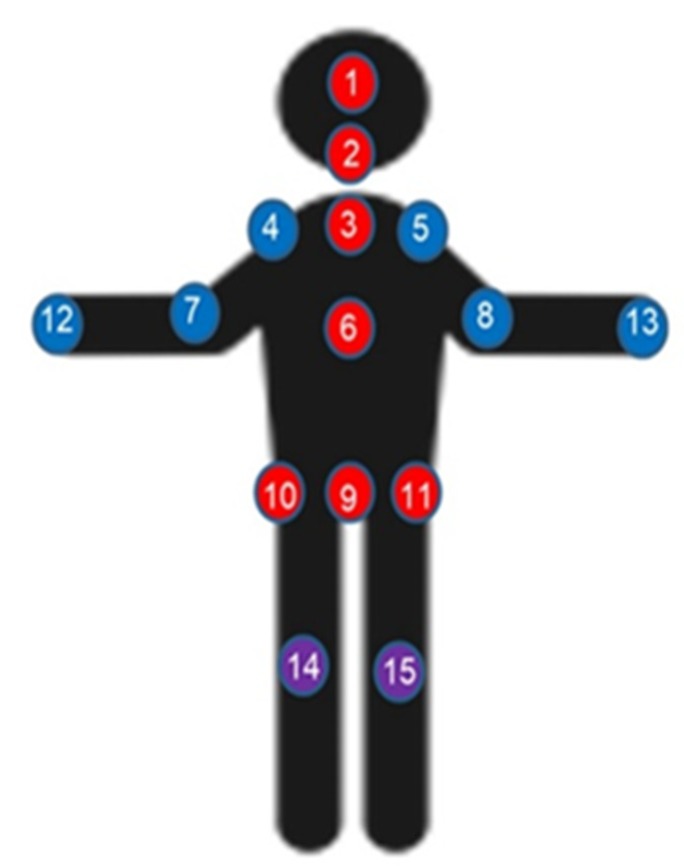
Joint annotation of the human body.

**Figure 2 sensors-20-01291-f002:**
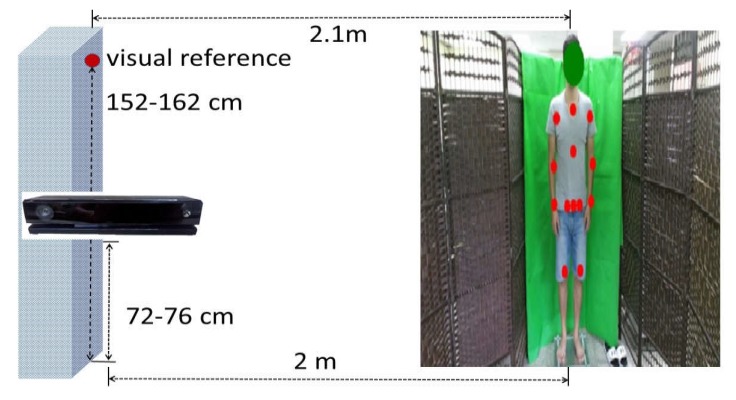
Experimental setup.

**Figure 3 sensors-20-01291-f003:**
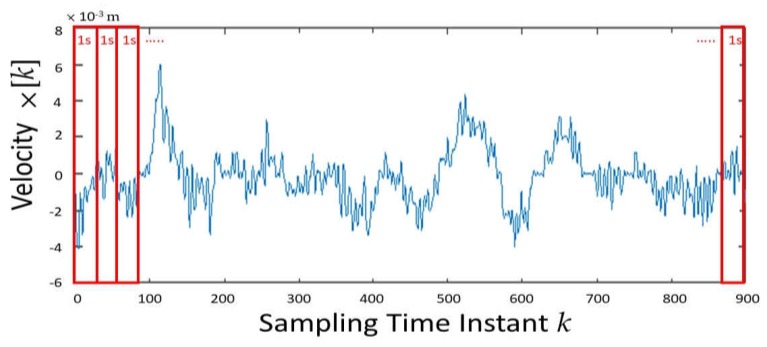
The 30-s time period of a joint velocity signal was divided into 30 one-sec nonoverlapping subintervals.

**Figure 4 sensors-20-01291-f004:**
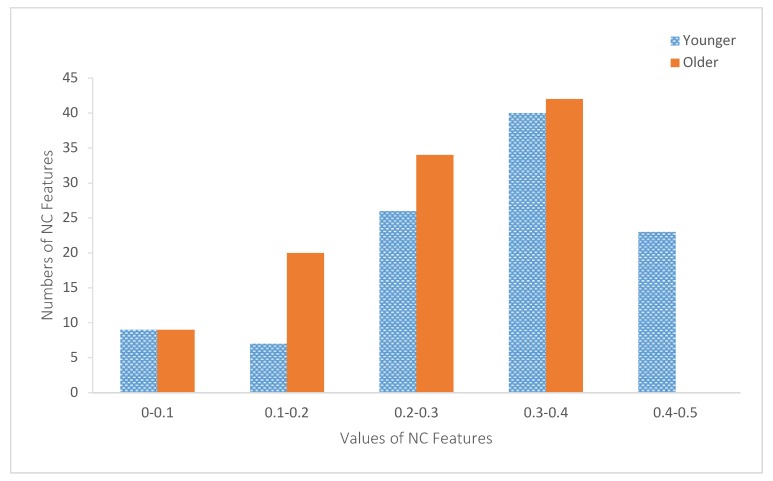
Histograms of the negative correlation (NC) features of both age groups.

**Figure 5 sensors-20-01291-f005:**
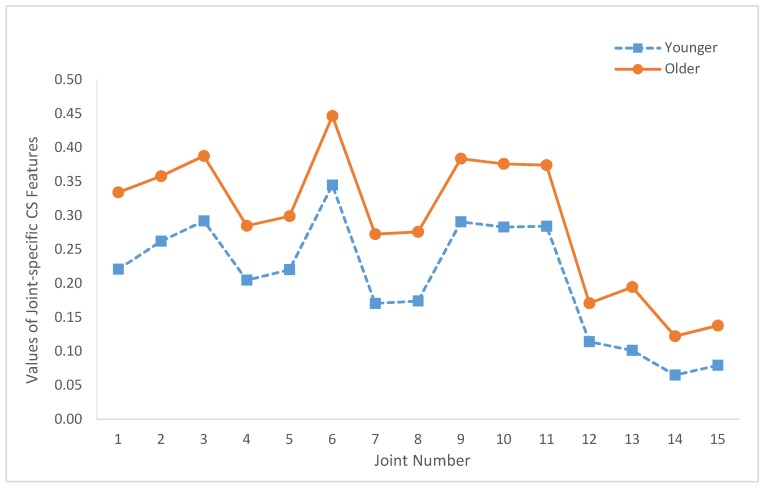
Summary of the joint-specific coupling strength (CS) features (JCS*_i_*, *i* = 1, …, 15) of both age groups.

**Figure 6 sensors-20-01291-f006:**
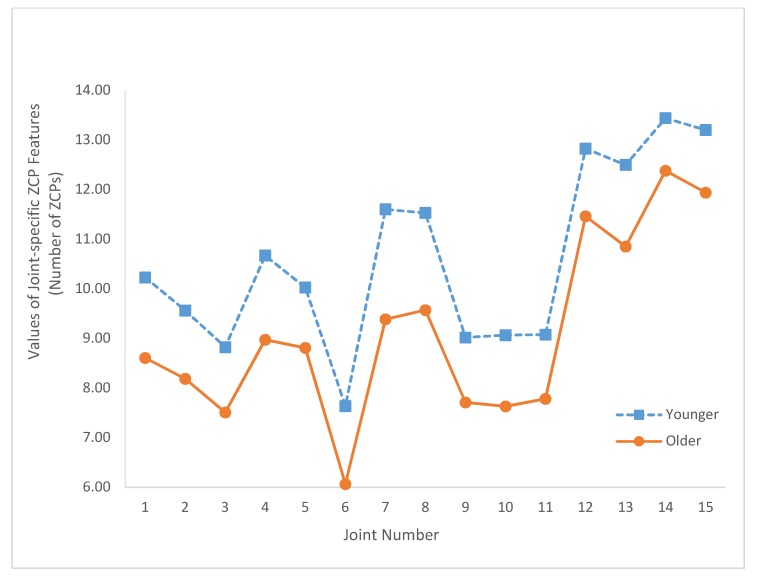
Summary of the joint-specific zero-crossing point (ZCP) features (JZCP*_i_*, *i* = 1, …, 15) for both age groups.

**Figure 7 sensors-20-01291-f007:**
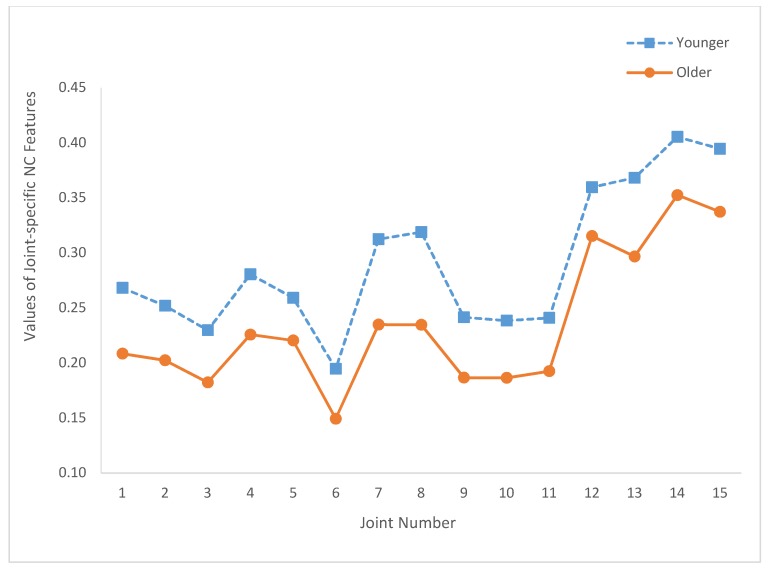
Summary of the joint-specific NC features (JNC*_i_*, *i* = 1, …, 15) of both age groups.

**Figure 8 sensors-20-01291-f008:**
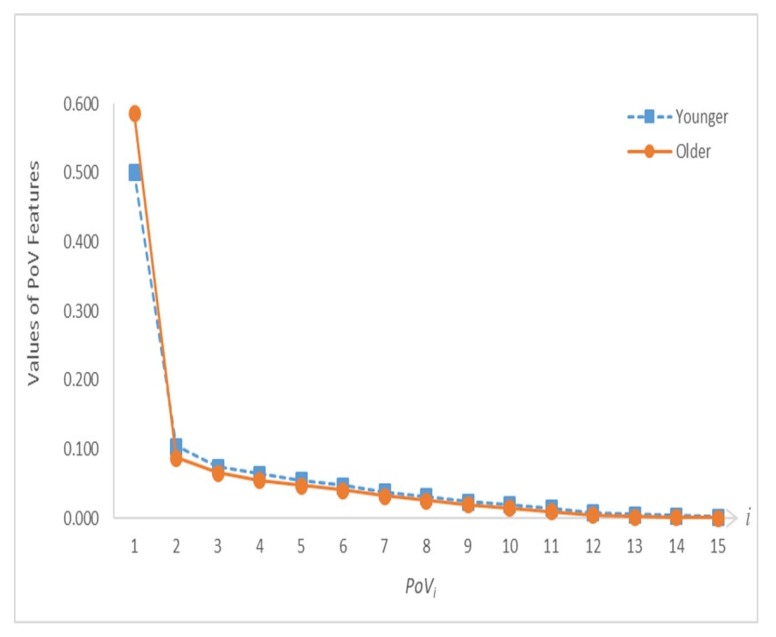
Proportion of variance (PoV) feature distribution curves of both age groups.

**Table 1 sensors-20-01291-t001:** Summary of specifications of depth sensor devices.

Devices	Depth Sensing	Color Images	Depth Image	FOV
Kinect V2	0.5–8 m	1920 × 1080@30fps	512 × 424@30fps	70° horizontal 60° vertical
Intel Realsense D415	0.3–10 m	1920 × 1080@30fps	1280 × 720@90fps	65° horizontal 40° vertical
ASUS Xtion Pro	0.8–3.5 m		VGA: 640 × 480@30fps QVGA:320 × 240@60fps	58° horizontal 45° vertical
ZED Stereo Camera	0.3–25 m	3840 × 1080@30fps 2560 × 720@60fps	WVGA:1344 × 376@ 100fps	90° horizontal 60° vertical
Creative Senz3D	0.2–1.5 m	1920 × 1080@30fps	VGA: 640 × 480@60fps	77° RGB, 85° IR depth
Orbecc Astra	0.6–8 m	640 × 480@30fps	640 × 480@30fps	60° horizontal 50° vertical
LIPSedge DL	0.1–8 m	1920 × 1080@30fps	VGA: 60 × 480@30fps	75° horizontal 58° vertical

Annotation: FOV = Field of View.

**Table 2 sensors-20-01291-t002:** Summary of the *p*-Values of joint-specific features.

Joint Number	*p*-values for Joint-Specific Features		
*i*	JCS*_i_*	JZCP*_i_*	JNC*_i_*
1	3.29 × 10^−4^	0.008	0.003
2	1.00 × 10^−3^	0.016	0.015
3	1.00 × 10^−3^	0.021	0.010
4	1.40 × 10^−2^	0.015	0.021
5	1.90 × 10^−2^	0.049	0.073
6	1.86 × 10^−4^	0.001	0.003
7	2.00 × 10^−3^	0.003	0.002
8	1.00 × 10^−3^	0.002	0.001
9	3.66 × 10^−4^	0.011	0.002
10	1.00 × 10^−3^	0.006	0.004
11	1.00 × 10^−3^	0.010	0.006
12	7.00 × 10^−2^	0.036	0.097
13	5.00 × 10^−3^	0.019	0.018
14	1.70 × 10^−2^	0.019	0.020
15	3.30 × 10^−2^	0.034	0.039

Annotation: JCS*_i_* = the mean of coupling strength features of joint-specific level for joint *i*, JZCP*_i_* = the mean of zero-crossing point features of joint-specific level for joint *i*, JNC*_i_* = the mean of negative correlation features of joint-specific level for joint *i*.

**Table 3 sensors-20-01291-t003:** Summary of the comparative results of the PoV features.

PoV Features	Means		*p*-Values
	Younger	Older	
PoV _1_	0.502	0.588	1.59 × 10^−2^
PoV _2_	0.105	0.088	4.07 × 10^−2^
PoV _3_	0.074	0.066	6.61 × 10^−2^
PoV _4_	0.064	0.055	1.84 × 10^−2^
PoV _5_	0.055	0.048	6.34 × 10^−2^
PoV _6_	0.048	0.041	6.09 × 10^−2^
PoV _7_	0.039	0.033	1.42 × 10^−1^
PoV _8_	0.032	0.026	8.28 × 10^−2^
PoV _9_	0.025	0.020	6.44 × 10^−2^
PoV _10_	0.020	0.015	1.76 × 10^−2^
PoV _11_	0.015	0.010	3.50 × 10^−3^
PoV _12_	0.008	0.005	3.00 × 10^−4^
PoV _13_	0.006	0.003	1.30 × 10^−5^
PoV _14_	0.004	0.002	1.30 × 10^−6^
PoV _15_	0.003	0.001	1.11 × 10^−8^
E_PoV_	2.602	2.245	3.75 × 10^−5^

Annotation: PoV = Proportion of variance.

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
