# Peer review of "Study of Postural Stability Features by Using Kinect Depth Sensors to Assess Body Joint Coordination Patterns"

_sensors, 2020, doi:10.3390/s20051291_

Round 1

Reviewer 1 Report

Title

Consider: “Study of Postural….”

Abstract

    - “..changing speed..” to “..changing velocity..”

    - Line 33: “..to be more joint dependent than joint independent..”, sounds redundant.

Experiments

    - Consider renaming this section as “Materials”

    - Consider including some features about the Kinetic sensor 

    - In 2.1 subsection you can comment that the joints are divided into two groups: axial and limb.  Is joint 1 really a joint?

Methodology

    - Units of the axis of Figure 3 are missing (Time in Hz?)

    - What kind of movements did the subjects?

    - Please consider including a mathematical definition of CS, ZCP and NC features.

     - How Fig. 3 was obtained?

     - How Fig. 5 to 7 were obtained from the correlation coefficients cij?

Results

      - Line 236: “…second part of the Section..”. Which section?

       - Related to Fig. 5 to 7, I don’t understand the horizontal axis, because you obtain features between two joints. Please explain how these figures were obtained.

       - If possible, could you comment on how this study can be useful to detect postural instabilities? In my opinion, this is the main concern.

Conclusion

      - Please cut in half this section.

      - Consider moving from line 315 to the discussion section and from lines 293 to 299 to the conclusion.

      - Thank for including the limitations of your work. In this sense, If possible I would suggest commenting on how this study can be useful to detect postural instabilities. In my opinion, this is the main concern.

Author Response

Response to Reviewer 1:

  • Comment

Title: Consider: “Study of Postural….”

Response:

In responding to this comment, the title of this paper has been changed from “Developing of Postural ….” to “Study of Postural …”

  • Comment:

Abstract: “..changing speed..” to “..changing velocity..”

Response:

The term “..changing speed..” has been replaced by “..changing velocity..”

  • Comment:

Abstract: Line 33: “..to be more joint dependent than joint independent..”, sounds redundant.

Response:

To eliminate the redundancy, the original sentence “.. to be more joint dependent than joint independent.” has been changed into “..to be more joint dependent.

  • Comment:

Experiments: Consider renaming this section as “Materials”

Response:

To more accurately represent the contents of Section 2, the title of Section 2 has been changed from “Experiments” to “Materials”

  • Comment:

Experiments: Consider including some features about the Kinetic sensor

Response:

To more thoroughly describe the Kinect V2’s specifications that are relevant to this study, for the first paragraph of Section 2.1, the sentence “The Microsoft Kinect V2 sensor, also known as the Xbox One Kinect, was used to acquire the displacements of joint centers with a sampling rate of 30 frames/sec.” has been replaced by the following sentences

The Microsoft Kinect V2 sensor, also known as the Xbox One Kinect, provides five video related data streams which include color (1920x1080@30Hz), infrared (512x424@30Hz), depth (512x424@30Hz) images as well as body index (512x424@30Hz) and the skeleton information for every tracked person (25 joint centers@30Hz). Note that the joint positions are provided at a resolution of 4 bytes per coordinate and hence 12 bytes per joint. The tracking volume of Kinect V2 is defined by the field of view (FOV, 70Ëš horizontally, 60Ëš vertically) and the range of depth sensing (0.5–4.5 meters).

  • Comment:

Experiments:  In 2.1 subsection you can comment that the joints are divided into two groups: axial and limb.

Response:

To address the reason why we divided the joints into axial and limb groups, the following sentences have been added to the fifth paragraph of Section 5.

The reason why for such a group division study is to investigate the potential associations between the relative locations of the joints with respect to central axis and the values of the proposed features in order to gain more insights about the IJCPs.

  • Comment:

Experiments: Is joint 1 really a joint?

Response:

Since “joint 1” represents the center of the head and can not perform any rotational movement, therefore the reviewer is correct in questioning is “joint 1” a real joint. To clarify the role of “joint 1”, the following sentences have been added to the end of the first paragraph of Section 2.1.

Note that “joint 1” represents the center of the head which can not perform rotational movement, therefore “joint 1” is not rigorously a kinematic joint. Considering the velocity of the head center is different from the velocities of other body joints and the potential role of head movement on postural stability, this work included head center in this study. However, to simplify the corresponding statements and discussions, this manuscript still refers head center as “joint 1”.

  • Comment:

Methodology: Units of the axis of Figure 3 are missing (Time in Hz?)

Response: We thank the reviewer for pointing out such errors. The unit for the vertical axis is meter. The unit for the horizontal axis is sampling instant. The revised manuscript has corrected these errors accordingly.

  • Comment:

Methodology: What kind of movements did the subjects?

Response: As the title of Section 2.3 indicates, this work performed standing still experiment. In specific, Section 2.3 states that “…the participants were instructed to look straight ahead at a visual reference and stand still (with arms at the side) in a comfortable stance for 40 s..”.  In fact, participants were asked to refrain from making any movement during the experiment.

  • Comment:

Methodology: Please consider including a mathematical definition of CS, ZCP and NC features.

Response: In responding to this comment, the following sentences have been added second last paragraph of Section 3.1 to mathematically clarify the definitions of the proposed features.

Mathematically, for a pair of joint velocity signals, its CS feature is defined as the mean of the corresponding temporal correlation coefficient signal, its ZCP feature is defined as the number of zero-crossing points of this temporal correlation coefficient signal and its NC feature is defined as the ratio of time that this temporal correlation signal has a negative value.

  • Comment:

Methodology: How Fig. 3 was obtained?

Response: The purpose of Figure 3 is to illustrate how we divided a joint velocity signal into 30 nonoverlapping subintervals in order to compute the temporal correlation coefficient signal between joint velocity signal pairs. To more clearly illustrate how Figure 3 was obtained, by combining with the second paragraph of Section 3.1 of the previous version of the manuscript, the first paragraph of Section 3.1 has been rewritten as:

As shown in Figure 3, as a pre-processing step for feature generation, each of the 30-s joint velocity signals was divided into 30 1-s nonoverlapping subintervals. With a 30 Hz sampling rate, each of such subintervals shown in Figure 3 consists of 30 sampling points. The temporal correlation coefficient cij[k] is then specified as the Pearson product-moment correlation coefficient between the kth subinterval of joint i and the kth subinterval of joint j for k =1, 2, ..., 30.

  • Comment:

Methodology: How Fig. 5 to 7 were obtained from the correlation coefficients cij?

Response: The problem addressed in this comment also appear in a later comment. Therefore, we will present our response to this comment later.

  • Comment:

Results: Line 236: “…second part of the Section..”. Which section?

Response: This is clearly our mistake. For the revised manuscript, the first sentence of the last paragraph of Section 4 has been revised as “The second part of this section….”. Therefore, it should be clear now that the section being referring to is Section 4.

  • Comment:

Results: - Related to Fig. 5 to 7, I don’t understand the horizontal axis, because you obtain features between two joints. Please explain how these figures were obtained.

Response: The concern of this comment is very similar to a pervious comment which stated that “How Fig. 5 to 7 were obtained from the correlation coefficients cij?”. Therefore, we will address these two comments together.

Figure 5 depicts the values of these joint-specific CS values JCSi for both age groups. As described in the last paragraph of Section 3.1, JCSi represents the mean of the CS features associated with joint i. For example, for joint 1, its JCS value (i.e., JCS1) is the mean of CS1,2, CS1,3, …, CS1,15. For joint 2, its JCS value (i.e., JCS2) is the mean of CS2,1, CS2,3, …, CS2,15. The remaining JCSi values can be calculated in a similar manner.

Similarly, as described in the last paragraph of Section 3.1, Figures 6 and 7 were obtained in the same manner as Figure 5. Specifically, the joint-specific CS value of joint i (i.e., the value of JCSi) of Figure 5 represents the mean of the CS features associated with joint i for i = 1, …, 15. Similarly, for i = 1, …, 15, the joint-specific ZCP value of joint i (i.e., the value of JZCPi) of Figure 6 represents the mean of the ZCP features associated with joint i and the joint-specific NC value of joint i (i.e., the value of JNCi) of Figure 7 represents the mean of the NC features associated with joint i.

To more clearly illustrate how Figures 5 to 7 were obtained, the last paragraph of Section 3.1 has been rewritten as:

The features CSij, ZCPij, and NCij characterize the coupling properties for joint pairs. To study the relative role of each joint in inter-joint coordination, this work extends the utility of these features by introducing three sets of joint-specific features. In specific, to extend CS features to a joint-specific level for joint i, we calculate the mean of CSij for j = 1, 2, ..., 15 and j ≠ i. By denoting this feature as JCSi, this joint-specific CS feature is used to quantify the overall coupling strength of joint i. In an identical manner, this work also extends the utility of ZCP and NC features and denote their joint-specific features as JZCPi and JNCi, respectively.

In addition, the captions of Figures 5 and 7 have also been revised. In specific, the original caption of Figure 5 “Summary of the joint-specific CS features for both age groups” has now been replaced by “Summary of the joint-specific CS features (JCSi, i = 1 ,.., 15) for both age groups”. Similar revisions have also been made for Figures 6 and 7.

Based on these newly added materials, we hope that it is clear that the horizonal axes of Figures 5 to 7 represent the numberings of the joints.

  • Comment:

Results: -If possible, could you comment on how this study can be useful to detect postural instabilities? In my opinion, this is the main concern.

Response: We agree with the reviewer that detecting postural instabilities is the main concern. We have to admit that our study cannot link the proposed features directly to postural instabilities. However, since it has been found that postural instabilities increase significantly with age and, as demonstrated by our experimental results, the proposed features can very effectively detect the aging effects on IJCPs, we believe that the proposed features hold the potential to detect postural instabilities caused by other postural stability impairing factors. Toward this goal, although not presented in this paper, our experimental results have found that the results obtained by the proposed features are significantly correlated with the conventional postural steadiness results obtained by force platform measurements. A possible future work is to repeat our experiment to a group of subjects with history of falls and a group of age-matched healthy control subjects.

Finally, in responding to this comment, the following sentences have been added at the end of Section 5.

The results presented in this study can not directly associate the proposed features with postural instability problems. However, considering the facts that postural instabilities increase significantly with age and the proposed features can very effectively detect the aging effects on IJCP, we believe that the potential of the proposed features in detecting postural instabilities warrant further study. In fact, although not presented in this work, our preliminary experimental results have shown that the results obtained by the proposed approach are in agreement with the conventional postural steadiness results obtained by force platform measurements.

  • Comment:

Conclusion: - Please cut in half this section.

Response: By removing two paragraphs and simplify the contents of the remaining paragraphs, the length of the Conclusion section has been considerably reduced. In specific, the length of the Conclusion section has been reduced from 29 lines to 16 lines.

  • Comment:

Conclusion: - Consider moving from line 315 to the discussion section and from lines 293 to 299 to the conclusion.

Response: Line 315 has been moved to the end of the second last paragraph of the Discussion section. In addition, lines 293 to 299 have also been moved to the Conclusion section.

  • Comment:

Conclusion: - Thank for including the limitations of your work. In this sense, If possible I would suggest commenting on how this study can be useful to detect postural instabilities. In my opinion, this is the main concern.

Response: By adding serval sentences at the end of Section 5, this concern has already been addressed in responding to a previous comment. In addition, the following sentence which appears in the final paragraph of the Conclusion section also addresses this concern.

To further test the efficacy of the proposed features for detecting postural instabilities, a possible future study is to use the proposed features to characterize the effects of balance impairing factors such as sensorimotor deficits.

Reviewer 2 Report

In this paper, three sets of features have been proposed (coupling strength, changing speed and complexity of the inter-joint couplings) to quantify body joint coordination.

The paper is well written and the message is clearbut some issues should beaddressed:

Some grammatical and tense errors throughout, please check the paper. As the authors have pointed, the reliability and validity of the Kinect sensor for human joint center measurements have been extensively tested and verified in many experimental studies. However, there are other devices that can be used for the same purpose. So it is recommended to add these devices in the introduction of the manuscript. It is said that ‘The Microsoft Kinect Software Development Kit (SDK) 2.0 was used to obtain the location of 15 human joint centers’. Why 15, and not more or less joints? Please add units in all graphs. In my opinion, 45 participants (only 15 youths and 30 elders) are too few to affirm the conclusions presented in the manuscript. The results would have been stronger if tests had been carried out with more participants.

Author Response

Response to Reviewer 2:

  • Comment:

Some grammatical and tense errors throughout, please check the paper.

Response:

Thank you very much for this suggestion. Before submitting the previous version of the manuscript, the paper had been proofread and edited by a native English speaker. However, in completing the final draft of the previous manuscript, we might had made several errors. Based on the suggestion of the reviewer, we have carefully checked the updated manuscript to correct grammatical and tense errors.

  • Comment:

As the authors have pointed, the reliability and validity of the Kinect sensor for human joint center measurements have been extensively tested and verified in many experimental studies. However, there are other devices that can be used for the same purpose. So it is recommended to add these devices in the introduction of the manuscript.

Response: It is very true that there are other similar devices. In instead of presenting the details of all these devices, the revised manuscript refers readers to the following paper (published in 2019) that comprehensively reviewed many Kinect alternative devices.

Clark, R. A.; Mentiplay, B. F.; Hough, E.; Pua Y. H. Three-dimensional cameras and skeleton pose tracking for physical function assessment: A review of uses, validity, current developments and Kinect alternatives. Gait & Posture 2019, 68, 193-200.

To further address this comment, we have surveyed many recent studies that used similar devices for human motion capture and analysis. Based on our survey results, the following sentences and table have been added to the end of Section 1.

We have also surveyed recent studies about the applications of depth camera for human motion capture and analysis. Based on the survey results, the following table summarizes the specifications of the most popular RGB-D sensors. Among all these popular RGB-D sensing devices, the Microsoft Kinect V2 sensor provides high resolution color images, large FOV areas, and powerful software development kits for skeleton and joint detection. In addition, as previously addressed in this paper, the validity and reliability of the Kinect sensor have also been extensively studied and verified. These are the reasons why we chose Kinect V2 as our measurement device for joint tracking.

Table 1. Summary of specifications of depth sensor devices

Devices

Depth sensing

Color images

Depth image

FOV

Kinect V2

0.5–8m

1920x1080@30fps

512x424@30fps

70Ëš horizontal 60Ëš vertical

Intel Realsense D415

0.3–10m

1920x1080@30fps

1280x720@90fps

65Ëš horizontal

40Ëš vertical

ASUS Xtion Pro

0.8–3.5m

VGA: 640x480@30fps
QVGA:320x240@60fps

58Ëš horizontal 45Ëš vertical

ZED Stereo Camera

0.3–25m

3840x1080@30fps

2560x720@60fps

WVGA:1344x376@ 100fps

90Ëš horizontal 60Ëš vertical

Creative Senz3D

0.2–1.5m

1920x1080@30fps

VGA: 640x480@60fps

77° RGB,

85° IR depth

Orbecc Astra

0.6–8m

640x480@30fps

640x480@30fps

60Ëš horizontal 50Ëš vertical

LIPSedge DL

0.1–8m

1920x1080@30fps

VGA: 640x480@30fps

75Ëš horizontal 58Ëš vertical

  • Comment:

It is said that ‘The Microsoft Kinect Software Development Kit (SDK) 2.0 was used to obtain the location of 15 human joint centers. Why 15, and not more or less joints? Please add units in all graphs.

Response: The number of skeleton joints that can be identified by Kinect is 25. By assuming the relative motions between wrist, hand and thumb centers of the same arm to be negligible when standing still, we excluded hand (4 joints) and thumb (2 joints) joints from our study. In addition, the foot (2 joints) and ankle (2 joints) joints were also excluded since these joints are relatively motionless when a person is standing still. We did not further reduce the number of joints since the relative motions of the remaining joints do not seem to be negligible.

To address this comment, the following sentences have been added to the first paragraph of Section 2.1.

In this work, the Microsoft Kinect Software Development Kit (SDK) 2.0 was used to obtain the location of 25 human joint centers. By assuming the relative motions between wrist, hand and thumb centers of the same arm to be negligible when standing sill, hand and thumb joints were excluded from this study. Since ankles and feet are relatively motionless when standing still, these joints were also not included in this study.

For the comment that we should add units in all graphs, we have added unit for the y axis of Figure 6. In specific, the unit for the y axis of Figure 6 is the number of ZCPs. The reason why that many graphs has no unit is that the corresponding variables are dimensionless. For example, the y axis of Figure 5 represents the mean of multiple correlation coefficients which by definition is dimensionless. The y axis of Figure 7 is the ratio of two variables that have same unit and is therefore dimensionless.

  • Comment:

In my opinion, 45 participants (only 15 youths and 30 elders) are too few to affirm the conclusions presented in the manuscript. The results would have been stronger if tests had been carried out with more participants

Response: We agree with the reviewer that increasing the number of participants can further affirm the conclusions presented in this study. Unfortunately, due to time limitation for submitting the revised manuscript, we did not have enough time to increase the number of test subjects. However, the following information may be valuable in affirming the validity of our results. First, as described in the third paragraph of Section 4, among the 105 comparative results for CS features, 102 of them achieved statistical significance. For the ZCP and NC features, 98 and 97 of the 105 test results achieved statistical significance. Altogether, among the 315 comparative results of the proposed CS, ZCP and NC features, 300 achieved statistical significances. With the vast majority of our statistical test results achieved statistical significance, these consistent results clearly support the effectiveness of the proposed approach.

Nevertheless, to address the limitation caused by the relatively small number of participants, the following sentence has been added to the second paragraph of Section 6.

The limitations of this study are as follows: First, the validity of the results can be further affirmed by increasing the number of participants.

Reviewer 3 Report

MAJOR COMMENTS

The errors of Kinect with respect to other instrumentation should be reported from data of literature. Especially at the light of the limit of not analysing AP-data due to the poor accuracy of Kinect in this direction. I would suggest to report the precision for all the three directions, explain the reasons for which AP-data are judged not accurate and the others enough accurate.

Explain the reason for which the sizes of the two samples of subjects were different each other (15 vs. 30)

There are a lot of analysed joint pairs (105 as detailed in line 214): there is the possibility of an inflaction of first type error in using t-test: authors should adopt a Bonferroni correction on the level of significance.

MINOR COMMENTS

Lines from 96 to 101 are useless, the structure of the paper is not unusual, so this part can be removed.

Specify the distance of visual reference

Line 191 – It is unclear how the filter at 5Hz may affect the vectors collection (authors wrote that “30-s signal length and a sampling frequency of 30 Hz, 900 x[k] vectors were collected”). Was the filter applied to these vectors?  

Author Response

Response to Reviewer 3:

  • Comment:

The errors of Kinect with respect to other instrumentation should be reported from data of literature. Especially at the light of the limit of not analysing AP-data due to the poor accuracy of Kinect in this direction. I would suggest to report the precision for all the three directions, explain the reasons for which AP-data are judged not accurate and the others enough accurate.

Response: The statement of our original manuscript about the relatively limited accuracy of Kinect sensor in AP direction was based on our own experimental results. In fact, we had tried to apply the proposed approach to the AP direction and failed to obtain satisfactory results. In responding to this comment, we have also reviewed many previous studies which tested the accuracy and validity of Kinect sensor. However, these studies only investigated the efficacy of Kinect senor in determining the displacement signals of the joint centers. In comparison, the features proposed in this work were all based the velocity signals of the joint centers. Consequently, the previous accuracy and validity test results for Kinect sensor are not necessarily applicable to this work.

The unsatisfactory results that we obtained for the AP-direction data may be related to the following property of the Kinect sensor. As described in the first paragraph of Section 2.1, compared to the 1920x1080 resolution of the color camera, the resolution of the depth image is only 512x424. As a result, with a relatively larger quantization error, the numerical differentiation error in the AP-direction is expected to be larger that of the ML-direction.

In responding to this comment, to be conservative, we remove the following sentence from the first paragraph of Section 2.4.

Second, AP direction data were not included because the AP direction measurement accuracy of Kinect is inferior to the accuracy of the other two directions.

and replace it with the following sentences:

Second, AP direction data were not included since our preliminary experimental results suggest that the proposed features are relatively ineffective in dealing with the AP direction velocity signals.

We want to thank the reviewer for this constructive comment, so that we can more conservatively and more accurately report why only ML direction data was studied in this work.

  • Comment:

Explain the reason for which the sizes of the two samples of subjects were different each other (15 vs. 30)

Response: The reason why the older age group has more participants (30 subjects) than the younger age group (15 subjects) is that aging is related to many postural stability impairing factors and each of which may have different impacts on inter-joint coordination patterns. In view of such uncertainties, we decided to have a larger sample size for the older age group. The other reason that we did not increase the size of the younger age group to further affirm our statistical test results is that we were able to obtain satisfactory and consistent test results with the current samples. For example, among the 45 hypothesis test results summarized in Table 2, 43 of them achieved statistical significance. Similarly, as described in the third paragraph of Section 4, among the 105 independent hypothesis tests for the CS features, 102 of them achieved statistical significance. Since increasing the number of test subjects would increase the cost and time of our experiments, with the support of the consistent statistical test results, we did not try to equal the size of the two samples by increasing the number of younger test subjects.

Based on the above reasons, to address the concern of this comment, the following sentences have added to the end of Section 2.2.

Aging has been known to be associated with many postural stability impairing factors each of which may have different impacts on IJCPs. In view of such uncertainties, the sample size of the older age group of this study was chosen to be larger than that of the younger age group.

  • Comment:

There are a lot of analysed joint pairs (105 as detailed in line 214): there is the possibility of an inflaction of first type error in using t-test: authors should adopt a Bonferroni correction on the level of significance.

Response: We thank the reviewer for this very valuable suggestion since we did not consider the possibility of using Bonferroni correction.

By adjusting the significance level, the Bonferroni correction has been used to reduce the risk of making a type I error when multiple statistical tests are being performed simultaneously. For example, as the reviewer pointed out, this study performed 105 hypothesis tests for the CS features. In this case, with a significance level of 0.05, the probability that concluding that a difference is presented when it is not is 1 – (1-0.05)105 ≈ 0.9954. Clearly, such a likelihood of making a type I error is not acceptable. To resolve this problem, with N denoting the number of hypothesis tests, Bonferroni correction changes the significance level from α to α/N. For our example, this correction reduces the probability of making a type I error to 1 - (1-0.05/105)105 ≈ 0.0488 which is apparently more appropriate.

Despite its widespread use, as addressed in a review paper by Armstrong (2004), there has been continuing controversy regarding the use of Bonferroni correction. One of the criticisms is that the test results obtained by Bonferroni correction is of little relevance to researchers who wish to assess the statistical significance of individual tests since one of underlying assumptions of Bonferroni correction is that the two groups being compared are identical in all comparisons. This is a serious drawback for our study since we want to independently test the efficacy of the proposed features on an individual basis. In addition, by using smaller significance level to reduce the probability of making a type I error, Bonferroni correction increases the probability of making a type II error and decreases the power of the hypothesis test. Therefore, the necessity of the Bonferroni correction depends on the circumstances of the study (Armstrong 2004).

Armstrong RA. When to use the Bonferroni correction. Ophthalmic Physiol Opt

2014;34:502-508.

Since the purpose of the Bonferroni correction is to limit the probability of making a type I error, whether we need to use Bonferroni correction depends on the type I error associated with our results. As presented in the third paragraph of Section 4, for the 105 hypothesis tests performed for the CS features, 102 of them achieved statistical significance. By assuming that the differences of these CS features between both age groups are all statistically insignificant, the probability of achieving 102 or more statistically significant test results can be readily determined from the following binomial distribution formula by assuming the significance level to be 0.05

which is extremely small. Note that we obtain similar negligible results for all the proposed features including ZCP, NC, JCS, JZCP and JNC.

It is true that we have conducted multiple hypothesis tests for our features. However, since the vast majority of these tests achieved statistical significance, as shown by the example shown above, type I error is very unlikely to occur in this study. Considering the side-effects of Bonferroni correction on the type II error and the hypothesis test power, we hope that the reviewer agrees that presenting our statistical results without using Bonferroni correction can more appropriately characterize the efficacy of the proposed approach.

  • Minor Comment:

Lines from 96 to 101 are useless, the structure of the paper is not unusual, so this part can be removed.

Response: We agree with the reviewer that these contents can be removed. Therefore, we have removed these lines from the manuscript.

  • Minor Comment:

Specify the distance of visual reference

Response: We thank the reviewer for remaining us to provide this information to the readers. In responding to this comment, we have added the following sentence to Section 2.3.

The distance between the visual reference and the test subject was about 2m.

  • Minor Comment:

Line 191 – It is unclear how the filter at 5Hz may affect the vectors collection (authors wrote that “30-s signal length and a sampling frequency of 30 Hz, 900 x[k] vectors were collected”). Was the filter applied to these vectors?

Response: The filter was designed to remove high frequency noise signals. Since the test subjects were asked to stand still in a comfortable stance during the experiments, there was no fast movement in any of the body joints. Based on the frequency spectra results, we discovered that the bandwidths of these joint center signals were all smaller than 5 Hz. Therefore, using a low-pass filter with a cutoff frequency of 5 Hz can remove unwanted noises without degrading the fidelity of the joint center signals.

The filer was indeed applied to the 30-s signal of x[k]. With the sampling frequency of 30 Hz, the dimension of these x[k] vector is 900. The lowpass filtering operation did not alter the dimension of x[k].

Round 2

Reviewer 1 Report

Thanks for considering my suggestions.

The paper is now better presented and it could be accepted.

Reviewer 2 Report

All comments raised by the reviewer have been nicely addressed.

Reviewer 3 Report

The authors have taken into account all my suggestions and the manuscript has now been improved